# Pediatric Myeloid Sarcoma, More than Just a Chloroma: A Review of Clinical Presentations, Significance, and Biology

**DOI:** 10.3390/cancers15051443

**Published:** 2023-02-24

**Authors:** Kristin E. Zorn, Ashley M. Cunningham, Alison E. Meyer, Karen Sue Carlson, Sridhar Rao

**Affiliations:** 1Department of Pediatrics, Division of Hematology/Oncology/Transplantation, Medical College of Wisconsin, Milwaukee, WI 53226, USA; 2Versiti Blood Research Institute, Milwaukee, WI 53226, USA; 3Department of Pathology, Medical College of Wisconsin, Milwaukee, WI 53226, USA; 4Department of Medicine, Division of Hematology/Oncology, Medical College of Wisconsin, Milwaukee, WI 53226, USA; 5Department of Cell Biology, Neurobiology, and Anatomy, Medical College of Wisconsin, Milwaukee, WI 53226, USA

**Keywords:** myeloid sarcoma, chloroma, acute myeloid leukemia, pediatric

## Abstract

**Simple Summary:**

Childhood acute myeloid leukemia (AML) remains a cancer with poor overall outcomes. Myeloid sarcomas (MS) are extramedullary masses of leukemia cells that can develop in patients with AML. In children, MS occurs more frequently than described in adults. Their clinical significance in both pediatric and adult patients with AML is unclear. In this review, we aim to summarize the current knowledge of MS in children and its underlying biology in the hopes of sparking future studies and ultimately improving treatment options for children with AML.

**Abstract:**

Myeloid sarcomas (MS), commonly referred to as chloromas, are extramedullary tumors of acute myeloid leukemia (AML) with varying incidence and influence on outcomes. Pediatric MS has both a higher incidence and unique clinical presentation, cytogenetic profile, and set of risk factors compared to adult patients. Optimal treatment remains undefined, yet allogeneic hematopoietic stem cell transplantation (allo-HSCT) and epigenetic reprogramming in children are potential therapies. Importantly, the biology of MS development is poorly understood; however, cell-cell interactions, epigenetic dysregulation, cytokine signaling, and angiogenesis all appear to play key roles. This review describes pediatric-specific MS literature and the current state of knowledge about the biological determinants that drive MS development. While the significance of MS remains controversial, the pediatric experience provides an opportunity to investigate mechanisms of disease development to improve patient outcomes. This brings the hope of better understanding MS as a distinct disease entity deserving directed therapeutic approaches.

## 1. Introduction

Myeloid sarcomas (MS) are extramedullary tumors of myeloid blasts forming masses disrupting normal tissue architecture in patients with acute myeloid leukemia (AML) [1,2,3]. They are also known as myeloblastomas, granulocytic sarcomas, chloroleukemia, and chloromas given the historically green appearance of the tumors resulting from myeloperoxidase exposure to air. Importantly, there is no clearly accepted definition of what qualifies as MS. Most agree that discrete tumor masses of myeloid blasts are MS; however, whether gingival infiltration and masses within lymph nodes, the liver, or spleen should also be considered MS is debated given their propensity for generalized infiltration. Central nervous system (CNS) leukemia is also a challenge with categorization including both cerebral spinal fluid (CSF)-positive disease and CNS infiltrates/masses on imaging. This has made clear, consistent reporting of clinical presentation and outcome data difficult given the lack of consensus within the literature. Given this limitation, the following terms will be used for this review: extramedullary disease and MS. Extramedullary disease will more broadly refer to leukemic disease outside the bone marrow/peripheral blood, while MS will be specific to myeloid blast tumors. These terms are not used interchangeably and are used as defined to more accurately portray the referenced literature, with extramedullary disease as an umbrella term that also includes MS.

MS most frequently presents with a mass in subcutaneous/soft tissue, bone, and skin (also known as leukemia cutis). Case reports include masses and infiltrative involvement in nearly every conceivable tissue including the GI tract, reproductive organs, CNS, heart, lungs, kidneys, and breast [4]. Interestingly, MS, while most often seen concurrently with intramedullary AML, can occur in isolation in the absence of bone marrow disease. MS can also occur in the setting of a preceding hematologic disease such as myelodysplastic syndrome (MDS) or myeloproliferative neoplasms (MPN). Finally, MS can develop as a relapse following a hematopoietic malignancy, including after allogeneic hematopoietic stem cell transplantation (allo-HSCT) [1,2].

Although AML is seen primarily in older adults with a median age at diagnosis of 68 years, AML accounts for 10 to 15% of acute leukemias in children [5,6]. Pediatric AML differs from AML in adults in terms of clinical course, outcomes, and genomic landscape [7,8,9]. MS in pediatrics represent an inadequately understood aspect of AML. MS presentations offer another distinction between pediatric and adult AML with opportunities for improvement in diagnosis, management, and further investigation into the biological mechanisms of development and treatment resistance.

This review will discuss the clinical presentations and reported outcomes of pediatric patients with MS including post-allo-HSCT, imaging approaches to diagnosis, and finally, the biology of MS will be addressed. While this review focuses on pediatric MS, important comparisons with adults will also be discussed.

## 2. Pediatric Clinical Presentation, Incidence, and Outcomes

Although generally considered a rare presentation, MS and extramedullary AML are common in children with AML. Numerous cooperative groups and large institution studies have reported both characteristics and outcomes associated with extramedullary disease (Table 1). The inconsistent terminology surrounding MS and extramedullary disease prevents direct comparison across these studies. Despite this limitation, these studies provide helpful data about the clinical features and associations seen as well as insight into outcomes.

### 2.1. Incidence

The incidence of MS varies widely, particularly when comparing adults and children. This is predominantly related to the lack of consistency in MS evaluation and reporting. There is no standard recommendation for patients with AML to undergo screening evaluation for MS and the true incidence is likely higher than that reported given the potential for asymptomatic occult tumors. Pediatric studies describe an incidence of MS ranging from 5.7% to as high as 40% with the expansive definition of extramedullary disease, although most commonly it is between 10 and 25% [10,13]. The distribution of common anatomic sites in children is illustrated in Figure 1. A lower incidence of MS is generally reported in adults (4–9%) with newly diagnosed AML, although this is likely an underestimate considering a recent prospective study [24]. It is unclear why such differences exist between children and adults, but this may be related to differences in diagnostic evaluations performed or inherent differences in the leukemias that children develop compared to adults in terms of mutational spectrum [7,8,9].

### 2.2. Clinical Associations

In children, MS is typically associated with a younger age at diagnosis, particularly in infants, and more frequently in males, with 55–75% of patients with MS being male [19,21,22]. Higher WBC counts and hepatosplenomegaly are often seen in the presence of MS, although less consistently. Additionally, the FAB M4 (acute myelomonocytic) and M5 (acute monocytic) subtypes are most commonly associated with MS. Cytogenetically, the most frequently described associations are inv(16), t(8;21), and chromosome 11 abnormalities, namely 11q23, with both deletions and rearrangements [12,15,17,18,19,26]. By contrast, in adults, a recent study of 1,583 patients identified increased odds of extramedullary disease in patients with *PTPN11*, *NPM1,* and *FLT3-ITD* mutations with no association with inv(16) or t(8;21) and decreased odds of extramedullary disease with *IDH2* and *CEBPA* mutations [27]. Pediatric AML has distinct mutational profiles in comparison to adult AML, and similarly, the AML genetic profile associated with pediatric MS appears distinct from that of adults. This suggests different biologic drivers of pediatric AML and MS or extramedullary disease development, and potential significance for treatment and outcomes.

### 2.3. Outcomes

There is no consensus on the influence of MS on prognosis in children with AML. Table 1 summarizes many of the larger studies, with notably conflicting results. Although typically considered a presentation of advanced disease, event free survival (EFS) and relapse free survival (RFS) do not consistently demonstrate worse outcomes for MS.

Reports from Children’s Oncology Group (COG) demonstrate an improved EFS in subsets of MS patients with non-skin MS, later defined as orbital MS and CNS MS, with otherwise similar outcomes to non-MS patients in those with other sites of MS disease including skin [12,15]. A single center report from India similarly describes improved EFS and overall survival (OS) in patients with MS excluding CSF-only disease [18].

By contrast, other studies describe no significant association between MS or more general extramedullary disease and EFS [11,16,17]. A Turkish single center study found significant effects on outcomes for patients with MS only when less intensive treatment was given [13]. The Japanese childhood AML cooperative study group also only found inferior EFS with extramedullary disease in the setting of WBC count >100 × 10^9^/L [14]. By contrast, other reports describe significantly worse EFS and OS for children with AML and MS compared to those without MS [10,19,25]. Collectively, this indicates that whether MS is a critical driver of outcomes remains an important unanswered question within the field.

Favorable cytogenetics, including core binding factor mutations, are common in patients with MS. In evaluation of children with low risk AML (e.g., inv(16), t(8;21), *NPM1* mutated without FLT3-ITD mutation, and *CEBPA* mutation), reports demonstrated worse RFS and EFS in patients with MS present compared to those without MS present [20,21,22]. This suggests that even in otherwise favorable AML, the presence of MS could be relevant for both prognosis and potentially risk stratification.

Although controversy remains in broadly assigning prognostic impact to the presence or absence of MS in pediatric leukemia, the significance should not be simply ignored. Particularly in patients with t(8;21), inv(16), or chromosome 11/*KMT2A* abnormalities, evaluating for the presence of MS may be significant in considering therapeutic approaches. Additional studies are needed to prospectively identify patients with MS, as defined by clear criteria, to determine the impact on prognosis, the potential need for altering risk stratification, and defining remission status. This will be important to identify which patients may benefit from specific or intensified therapy regimens to improve outcomes.

## 3. Significance of Extramedullary Disease and Myeloid Sarcomas Post-Allogeneic Hematopoietic Stem Cell Transplant

Relapse of AML remains the predominant cause of treatment failure and death with allo-HSCT as the only curative option for many patients. The role of allo-HSCT in the setting of MS is a moving target in children. However, new data are emerging to address this important point because anecdotal evidence suggests isolated MS relapse, in the absence of bone marrow relapse, is a common occurrence post-allo-HSCT.

The Japan Society for Hematopoietic Cell Transplantation (JSHCT) used their national database to identify pediatric AML patients that underwent allo-HSCT and found that the presence of extramedullary disease (both CNS disease and MS) had no impact on OS or leukemia-free survival (LFS) after transplant. However, the patients with extramedullary disease prior to transplant were more likely to have extramedullary relapses after transplant, with 41% of relapses being extramedullary. In comparison, those without prior extramedullary disease had only 6% extramedullary relapse, although the overall rates of recurrence were the same between the two groups [28]. Relapse with isolated MS was not separated out as a group and was therefore difficult to directly assess.

The Turkish Pediatric Bone Marrow Transplantation Registry recently reported on their experience with isolated extramedullary relapse (iEMR) in children following allo-HSCT, although they included both acute lymphoblastic leukemia (ALL) and AML. They found different risk factors for medullary relapse post-allo-HSCT versus iEMR. Transplant in CR2 or later or active disease at time of transplant and matched sibling donor transplants were independently associated with increased risk of medullary relapse as well as iEMR. The presence of chronic graft versus host disease (cGVHD) was conversely associated with decreased risk of medullary relapse with no impact on the risk of iEMR. iEMR rates were, however, independently higher in those with prior extramedullary disease [29]. A higher rate of second iEMR was also seen following a first iEMR at 58.8% versus after a first medullary relapse at 13% [29]. Local radiotherapy of extramedullary disease sites prior to transplantation and the presence of cGVHD had no impact on post-allo-HSCT iEMR, while cGVHD was protective in preventing medullary relapse [29]. This suggests that although a graft versus leukemia effect is helpful in preventing medullary relapse, this immune-mediated mechanism is not effective against MS masses and extramedullary sites of leukemia. A single site report from the University of Michigan found that children were also three times more likely than adults to experience an extramedullary relapse with an associated higher pretransplant extramedullary disease incidence [30]. The significance of extramedullary relapse, particularly in these settings, provides insight into mechanisms of disease resistance specific to the MS phenotype following allo-HSCT. Despite these concerns, allo-HSCT remains the best disease management for patients with high-risk AML.

Extramedullary disease prior to transplant is consistently associated with increased risk of extramedullary relapse after HSCT in both children and adults [30,31]. The presence of prior extramedullary disease (including CNS disease and MS) in adults with AML was not found to be an independent risk factor for post-allo-HSCT relapse, DFS, or OS in both a large CIBMTR analysis and a Canadian report [32,33]. This confirms the anecdotal clinical concern that allo-HSCT is more effective for medullary versus extramedullary disease.

Adults with iEMR post-allo-HSCT are more likely to have had prior extramedullary disease and GVHD present compared to those with medullary relapse [30,31,34,35]. Additionally, extramedullary relapse has a higher incidence following allo-HSCT than intensive chemotherapy alone [36]. These iEMRs may represent sanctuary sites in which immune-based therapies may be less effective and may result from a different mechanism of pathogenesis compared to medullary relapse. There is currently no treatment consensus for iEMR post-allo-HSCT, with a range of treatment approaches taken including local radiotherapy and systemic chemotherapy [37,38,39,40]. The significance of iEMR on outcomes compared to medullary relapse is more controversial, with conflicting studies limited by inclusion of both ALL and AML patients with known differences in the efficacy of graft versus leukemia effect between the two diseases [31,34]. One retrospective adult study, however, did report that allo-HSCT was an effective treatment for patients with MS compared to chemotherapy alone, although lack of complete MS remission prior to transplant had independently worse OS and PFS [41].

In summary, the presence of known or occult MS prior to allo-HSCT may be clinically important for a subset of patients, although additional studies are needed to define which groups may benefit. Furthermore, identifying the increased risk for iEMR following allo-HSCT can inform evaluations and management of patients during their post-transplant course. This emphasizes the importance of identifying and following MS in patients with AML prior to allo-HSCT and remaining vigilant to the possibility of iEMR.

## 4. Imaging Evaluation of Myeloid Sarcomas

The use of imaging to identify occult MS as well as re-evaluation of disease presence has remained inconsistent, both in frequency and modality, particularly in children. Ultrasounds of MS lesions typically show homogenously hypoechoic lesions with hypervascularity [42]. Computed tomography (CT) scans identify MS as isodense lesions compared to muscle with moderate enhancement with IV contrast media. Enhancement is more commonly homogenous (65%) versus inhomogenous (35%) [43]. MRI scans demonstrate predominantly T2 hyperintense (82%) or isointense (18%) lesions compared to muscle and T1 isointense (61%) or hypointense (39%) lesions with homogenous contrast enhancement and a mean apparent diffusion coefficient (ADC) on diffusion weighted imaging (DWI) of 0.57 × 10^−3^ mm^2^/s [43].

Fluorodeoxyglucose (FDG)-positron emission tomography (PET) scans are increasingly being used for diagnosing extramedullary disease, with MS lesions displaying moderate uptake of FDG [44]. A retrospective study including pediatric patients showed a sensitivity of 93% and a specificity of 71.4% limited by difficulty differentiating extramedullary leukemia disease from infectious/inflammatory entities [45]. The recent PETAML trial however prospectively evaluated adult patients with AML prior to therapy initiation with total body ^18^FDG PET/CT scans to determine prevalence of extramedullary disease. This showed a prevalence of 22% with a sensitivity of 77% and specificity of 97% [24]. Interestingly, leukemia cutis and CNS meningeal involvement were not necessarily ^18^FDG-PET-avid [24,45]. In addition, there were four patients who remained with residual ^18^FDG-PET-positive lesions despite complete marrow remission. Three of those four subsequently relapsed, suggesting there may be a specific role for ^18^FDG-PET imaging for remission evaluation of patients with AML [24]. Consideration should be given to prospectively evaluating patients with AML to identify MS lesions requiring focused follow-up and possible treatment modifications and may be of value in designing de novo AML clinical trials, particularly in pediatrics with a high incidence of MS.

## 5. Pathology of Myeloid Sarcomas

MS are infiltrative tumor masses of myeloid blasts that efface or disrupt the normal architecture of the involved organ. The leukemic blasts found in MS have heterogeneous morphology; however, monocytic differentiation is common where the blasts will show either myelomonocytic or monoblastic morphology [46]. An example of the histology of MS is shown in Figure 2. Immunophenotypic profiling by flow cytometry or immunohistochemical stains is often necessary for a definitive diagnosis, as many of these tumor masses may resemble carcinoma. Evaluation of markers of immaturity, including CD34 and CD117, are helpful in addition to other markers which are variably expressed on myeloid blasts including CD13, CD33, CD68 (KP1), CD45, and myeloperoxidase (MPO). In the setting of monocytic differentiation, expression of monocytic markers such as CD68, CD163, CD14, and/or non-specific esterase (NSE) can be seen.

Several translational studies have begun to investigate the genetic landscape of MS lesions in isolation as well as in comparison to their paired intramedullary leukemia counterparts. One recent study evaluated 7 adult trios of AML, MS, and normal tissue using capture-based next generation sequencing (NGS) of 479 cancer genes. Genes recurrently altered in these patients included *KMT2A*, *FLT3*, *NRAS*, *CEBPA*, *TP53*, *WT1*, and *NPM1*, with 84% of variants found in the AML also present in the MS [47]. Three of the seven patients had additional variants detected in the MS compared to the AML including additional *FLT3*, *SETD2*, and *NF1* mutations in the MS, while two had additional variants of *U2AF1* and *RAD21* in the AML but not the MS [47]. In the relapsed MS samples, there were increased single nucleotide variants (SNV) in the MS [47]. Another study evaluated 6 isolated MS tumors (without concurrent AML) and performed a 21 gene targeted panel of AML and MDS associated genes. They found recurrent variants in the genes for *FLT3* (50%), *NPM1* (33%), and *KIT* (67%) and additional variants in *WT1*, *SF3B1*, *EZH2*, *ASXL1*, and *TET2* in one MS each [48]. The genomic reports of patient-derived MS are all limited by targeted NGS sequencing without exploration of novel gene variants that may be specific to MS pathogenesis. Additionally, RNA transcriptome analysis of MS is lacking in the literature and provides an opportunity for investigation of transcriptome-based changes that may contribute to MS development outside of genetic mutations. Such studies may also facilitate the identification of cryptic translocations, which are common in pediatric AML [7]. Although, typically, the genomic profile of MS is in concordance with the AML and marrow, this is not always true. Particularly in cases of isolated MS, NGS and molecular evaluation may inform targeted treatment options and should be included in diagnostic evaluation of patients.

## 6. Biological Understandings of Pathogenesis

The biology underlying development of MS remains poorly defined with no clear molecular determinants. Biological features such as cytogenetic changes, molecular abnormalities, and cell surface marker expression are not consistent across studies. Much of the work on MS development surrounds the invasiveness of AML as studied using in vitro transwell assays and infiltration in the spleen and liver. The simple infiltration of hematopoietic organs, however, appears distinct from MS development in non-hematopoietic sites with no clear biological explanation. The development of MS appears to require leukemia mobilization/release from the marrow environment, tissue invasion, and further changes leading to a tumor/mass phenotype, as illustrated in Figure 3. These steps will be further discussed below.

### 6.1. CXCR4

CXCR4 (CXC chemokine receptor 4, CD184) is the receptor for the chemokine matrix cell derivative-1 (SDF-1/CXCL12) and is expressed by most tissues as well as hematopoietic stem cells and leukemic blasts wherein it facilitates the retention of hematopoietic stem cells in the bone marrow niche [49,50]. The CXCR4/SDF-1 axis may contribute to chemoresistance through downstream signaling cascade dysregulation within leukemia cells [49,51]. Higher *CXCR4* expression has been seen in AML patients with extramedullary infiltration at diagnosis and extramedullary infiltration in childhood ALL [49,52]. The proposed mechanism of extramedullary involvement in acute leukemias is altered bone marrow homing and increased peripheral blood dissemination via a chemotactic gradient of SDF-1 with increased CXCR4 expression on the leukemia cells [53]. CXCR4/SDF-1 can promote the retention of AML cells within the skin of children with AML; however, CXCR4 expression by peripheral blood blasts was no different in patients with or without skin involvement [54]. Furthermore, a lack of association between SDF-1 polymorphisms and MS implies that small variants do not contribute to extramedullary disease development, although these have been previously of interest and described [55]. More common in adults, *NPM1*-mutated AML is associated with extramedullary disease and is associated with downregulation of *CXCL12* and *CXCR4* gene pathways [50].

While CXCR4 expression and signaling may be a contributing factor to extramedullary disease, its impact appears limited to initial release and migration of leukemia cells from the marrow and is not specific to MS development. Further work is needed to better characterize this mechanism and whether or how CXCR4 is contributing to discrete MS formation.

### 6.2. CD56

CD56 (also known as neural cell adhesion molecule-1 or NCAM1) is normally expressed by natural killer (NK) cells and other immune cell subtypes and is housed on chromosome 11q23.1. It is frequently described as part of the immunophenotype of AML with MS [1,56,57]. Expression patterns of CD56 are not consistently described, however, and in a population of adult t(8;21) AML patients, there was no association between CD56 expression and presence of extramedullary disease [58]. AML in adults with CD56 positivity is more commonly associated with worse 5-year EFS and OS; however, a report in low-risk patients shows no association with outcome [21,25,59]. Additionally, post-allo-HSCT CD56 positivity is not associated with extramedullary relapse [30]. Despite the frequent CD56+ immunophenotype, there is no described mechanism or in vitro data to suggest the significance of this finding. Additional experimental studies are required to determine if CD56 is simply a biomarker of MS or is required for MS development.

### 6.3. Integrins and Cell Adhesion Molecules (CAMs)

An AML-extracellular matrix interaction is likely critical to the development of MS. This is illustrated in transcriptome analysis of adult patient-derived AMLs demonstrating enrichment of cell surface gene sets in those AMLs with concomitant MS [60]. This includes integrin-α7 *(ITGA7*), which showed a higher expression in AML with associated MS in addition to high expression in MS samples [60]. Laminin 211 is a specific ligand of integrin-α7 that signals through the ERK signaling cascade [60]. While there is much described about the role of integrins and selectins in migration and homing of hematopoietic stem cells, there remains no clear mechanism by which these molecules facilitate MS formation [61,62]. Further study is needed to better evaluate the role of cell adhesion molecules in MS development and whether targeting these cell interactions may provide therapeutic benefit for patients with MS.

### 6.4. Vascular Endothelial Growth Factor (VEGF) and Receptor (VEGFR)

Angiogenesis plays a notable role in acute leukemia with increased microvascular density in AML and adult MS [63]. VEGFR2, the major mediator of the mitogenic, angiogenic, and permeability effects of VEGF, may contribute to the development of MS [64]. VEGF signaling via the PI3K/Akt pathway in the setting of *hERG1* expression was necessary for an in vitro migratory phenotype in AML cells [65]. In adults, the small molecule VEGFR2 tyrosine kinase inhibitor apatinib (also known as TN968D1) demonstrated enhanced antileukemic effects in ex vivo cytotoxicity studies from patient-derived AML samples with associated extramedullary disease [66]. Angiogenesis is well-described in the pathogenesis of other malignancies and it is reasonable to think that a unique perturbation may play a role in the migration or tumor formation of MS.

### 6.5. Matrix Metalloproteinases (MMP)

In vitro studies have described the role of MMP secretion (MMP-2 and MMP-9) by leukemia cells contributing to invasion capacity, most notably of the blood-brain barrier, with upstream regulation by mitogen-activated protein kinases (MAPKs) and phosphoinositide 3-kinase (PI3-K)/AKT pathways [67,68]. Additionally, TIMP-2 (tissue inhibitor of metalloproteinase 2) upregulation has been seen with increased leukemia cell line (i.e., SHI-1) invasion both in vitro and in vivo with more extensive and severe extramedullary infiltration through both MMP-2-dependent and independent activities [69,70]. Other in vitro studies propose a role for the β2 integrin-proMMP-9 complex in the extramedullary phenotype of AML [71]. Type IV collagenase secretion enhanced by TNFα and TGFβ from a patient-derived MS cell line increased in vitro cell invasion with collagenase secretion demonstrated in the MS AML cell line but not other leukemia cell lines [72]. While tissue invasion by leukemia cells is likely required for MS development, not all of the critical players have been identified.

### 6.6. Epigenetic Dysregulation

Epigenetic dysregulation has been reported in the context of extramedullary disease and infiltration in AML. Enhancer of zeste homolog 2 (EZH2) is a histone methyltransferase and is the catalytic subunit of the Polycomb Repressive Complex 2 (PRC2), which deposits Histone 3 Lysine 27 trimethylation (H3K27me3). High *EZH2* expression is correlated with higher peripheral blood blast percentages as well as extramedullary infiltration in patients with AML with numerous well-established biological roles. In vitro studies suggest that migration of AML cells appears to be regulated by EZH2/p-ERK/p-cmyc/MMP-2 and E-cadherin signaling pathways [73]. *EZH2* is a frequently mutated gene in AML; however, *EZH2* has a variety of biologic influences and a unique role in MS formation remains undefined [7,9].

Altered DNA methylation is another described mechanism in the development of extramedullary disease with key enzymes frequently mutated in AML. DNA methyltransferase 3A (*DNMT3A*) mutations contribute to altered DNA methylation, subsequently resulting in increased expression of a subset of genes with specific roles in myeloproliferation and extramedullary hematopoiesis [74]. *DNMT3A* mutation appears to contribute to extramedullary CNS infiltration mediated by overexpression of *TWIST1*, a key epithelial mesenchymal transition transcription factor, which is not otherwise well described in AML [75]. Furthermore, *TET2* is a member of the ten-eleven translocation (TET) gene family and is a key enzyme for DNA demethylation and a critical regulator for hematopoietic stem cell homeostasis. Models using *TET2*-deficient mice demonstrated not only high incidence of MS development but also transplant ability of the MS cells as well as an in vivo response to azacitidine treatment [76]. Decreased *TET2* expression was also seen in patient-derived MS samples with further suggestion of methylation changes impacting MS development [77].

AML has many examples of mutations in epigenetic pathways that are enriched in AML more than many other disease entities and may not be directly related to their involvement in MS [9]. The role of epigenetic dysregulation in leukemia migration and invasion with described MS phenotypes is intriguing yet requires further study. Additional research may uncover future targetable pathways for MS treatment, and as noted below, case reports have demonstrated the safety and efficacy of hypomethylating agent use for patients with MS.

### 6.7. Other Biological Associations

Mesothelin (MSLN) is a cell surface protein hypothesized to be involved in cell adhesion and is overexpressed in a subset of AML patients. MSLN overexpression was strongly associated with KMT2A-R, t(8;21), and inv(16) as well as the presence of extramedullary disease in children and young adults with AML. Methylation profiling further demonstrated an inverse association between MSLN promoter methylation and MSLN expression, suggesting another impact of epigenetic dysregulation [78].

Versican (VCAN) overexpression in the setting of *NPM1*-mutated AML is associated with an invasive phenotype and higher expression levels in patients with skin infiltration [79]. Lysyl oxidase (LOX), which has roles in pediatric acute megakaryoblastic leukemia and in the creation of a growth permissive fibrotic microenvironment, was associated with increased extramedullary disease in adults with AML and high plasma LOX activity [80]. *WT1* overexpression has also been described in MS cases as well as in extramedullary relapsed disease [81,82]. *ERG* transcription factor overexpression, similar to that of Ewing sarcoma, has been seen in patient-derived MS samples [83]. Multiple studies describe other associations observed in AML and extramedullary disease, including increased expression of amyloid precursor protein (APP) in *AML1/ETO* leukemia cells perhaps mediating the p-ERK/c-Myc/MMP-2 pathway, expression of miR-29c&b2, circular RNA expression patterns, and expression of CD25 and CD117 [84,85,86,87,88,89,90]. Polo-like kinase 1 (PLK1), which is involved in cell cycle control, was effectively inhibited in vivo using a patient-derived leukemia in mice with improvement in extramedullary disease [91]. PD-1 and PD-L1 have been investigated given the described efficacy of checkpoint inhibitors; however, there was no difference in expression of *PD-1*/*PD-L1* in MS tested, and they may instead have more impact in the surrounding tumor microenvironment [92,93]. Using mouse models, others observe a maturation plasticity of leukemia cells, with potential implications for chemotherapy resistance as a mechanism for extramedullary relapse [94]. A *PIM2/MYC* co-expressed mouse model demonstrated consistent and lethal in vivo MS development with *MYC* expression likely contributing to the phenotype [95]. Mouse models have also demonstrated cooperation between *MLL/AF10* and activating *KRAS* mutations, with increased cell adhesion properties contributing to in vivo MS formation via *Adgra3* and *Hoxa11* [96,97].

While many different mechanisms have been suggested in the development of MS, there remains no clear understanding of the pathogenesis. As such, it is hard to definitively identify the potential molecular determinants causing MS formation is some AML patients but not others. While the pathogenesis remains to be fully elucidated, prior studies suggest that there are likely multiple steps leading to MS development, including release from the bone marrow (which may be represented by higher WBC counts associated with MS), tissue invasion, and discrete mass formation, with the latter being the most consequential with regards to leukemia and the least described. Investigating how these different steps may cooperate and ultimately how the leukemia cells aggregate and sustain an aggregated phenotype requires dedicated study. Furthermore, the immune evasive or immunosuppressive microenvironment of MS illustrated in the post-allo-HSCT setting highlights that there is much more to learn about the pathogenesis of MS and its uniqueness with respect to its systemic/intramedullary AML counterpart.

## 7. Treatment Considerations

There is no consensus on the best treatment approach for management of MS, particularly isolated extramedullary disease. In pediatrics, systemic chemotherapy has been favored with consideration of allo-HSCT independent of the presence of MS.

The general approach to management in pediatric patients has evolved over the last three decades. In prior large cooperative group study treatment protocols (e.g., CCG 2961), children with MS would receive radiation therapy to the affected sites following initial induction chemotherapy, given MS responsiveness to irradiation. Although part of protocol therapy, many patients were not irradiated and outcomes demonstrated no difference in 5-year EFS, similar to smaller cohorts [12,98]. In adults, radiation therapy is more commonly used to treat isolated relapses, but the effects are not typically sustained and both localized and medullary relapse following radiotherapy are common [30,99]. Although prior COG studies included radiotherapy for treatment of MS sarcomas, this is no longer standard of care in the US; however, it continues to be recommended for MS in Berlin–Frankfurt–Munster (BFM) studies [17,100].

Given the epigenetic basis of AML development, inclusion of novel epigenetic approaches in treatment are increasing in utility for AML [9,101,102]. The hypomethylating agents azacitidine and decitabine have demonstrated efficacy in management of extramedullary disease in AML, including in pediatric patients [103]. Case reports in pediatrics have demonstrated complete response to monotherapy with azacitidine in MDS patients who received allo-HSCT as consolidation [103]. In the setting of post-allo-HSCT relapses of MS, multiple case reports in adults demonstrate efficacy of azacitidine or decitabine including complete response [104,105]. Hypomethylating agents in adults with AML and extramedullary disease showed improvement after one–two cycles and complete or near complete resolution of MS following four–five cycles [106,107,108,109]. Venetoclax has also shown activity against MS [109,110,111]. The utility of venetoclax and hypomethylating agents suggests a role for epigenetic reprogramming as a means for MS treatment, although DNA methylation-based mutations including *DNMT3A*, *IDH1*, *IDH2,* and *TET2* are far less common in children than in adults and translation of utility is more challenging [9,112]. While no consensus exists regarding optimal treatment of MS, particularly in pediatrics, radiation therapy is unlikely to contribute to durable remission of disease and expanding chemotherapeutic options to include hypomethylating agents and venetoclax in both initial chemotherapy regimens or as maintenance therapy following allo-HSCT should be considered and deserves further investigation.

## 8. Conclusions: Knowledge Gaps and Areas for Improvement

Children with AML and MS are distinct from adults. Given these differences, it is necessary to further study MS in the context of the driver lesions specific to children. MS remains a known clinical presentation with unclear impact on prognosis, risk stratification, and potential consequences in the setting of allo-HSCT. The advancement of imaging techniques and data for MS provides the opportunity for more directed and prospective evaluation in children. Collectively, this highlights the need for further large-scale cooperative group studies with clear criteria for the identification of MS in children.

While there is no specific treatment approach for children with MS, the use of intensive systemic chemotherapy remains at the forefront. However, additional studies are required to determine if epigenetic or immuno-oncology therapies may be beneficial. The role of allo-HSCT continues to be important as a curative option for many patients with high-risk AML; although, considering its potential lack of efficacy in the setting of extramedullary disease, the role of an immunosuppressive microenvironment in MS requires additional study. Further investigation into the potentially immunosuppressive MS microenvironment will be crucial to improving efficacy of allo-HSCT and managing isolated extramedullary relapses post-HSCT.

Although many different biological associations exist, there is an ongoing lack of clarity as to how leukemic blasts can not just invade tissues but form discrete tumors. Experiments delineating the potential epigenetic and transcriptomic differences between medullary AML disease and MS are required to identify the underlying molecular mechanisms that promote MS development. Understanding how leukemic blasts transform into and sustain an MS phenotype is critical to identifying specific targetable mechanisms. Understanding and combatting chemotherapy resistance and immune escape will ultimately improve survival in patients with AML and MS.

## Figures and Tables

**Figure 1 cancers-15-01443-f001:**
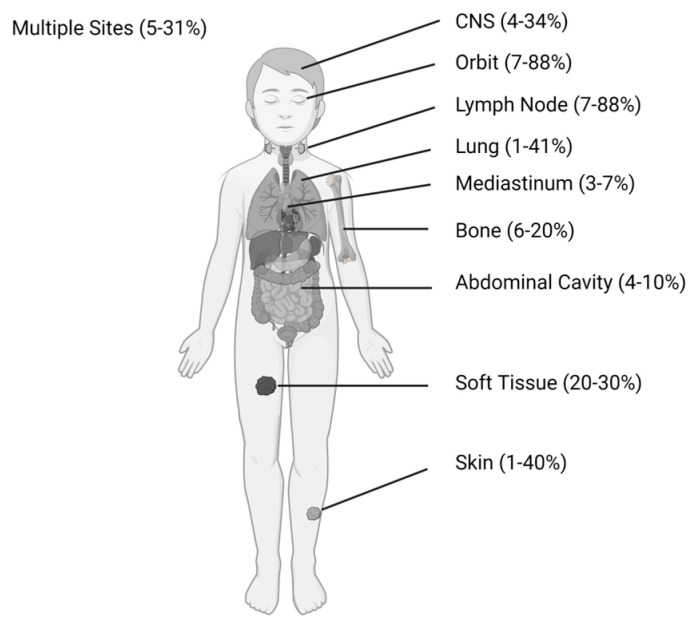
Common locations of initial presentation of MS in children with percentage of MS site involvement. Abbreviations: CNS, central nervous system; MS, myeloid sarcoma [11,12,13,14,15,16,17,18,21,22,23,25]. Figure created with BioRender.com.

**Figure 2 cancers-15-01443-f002:**
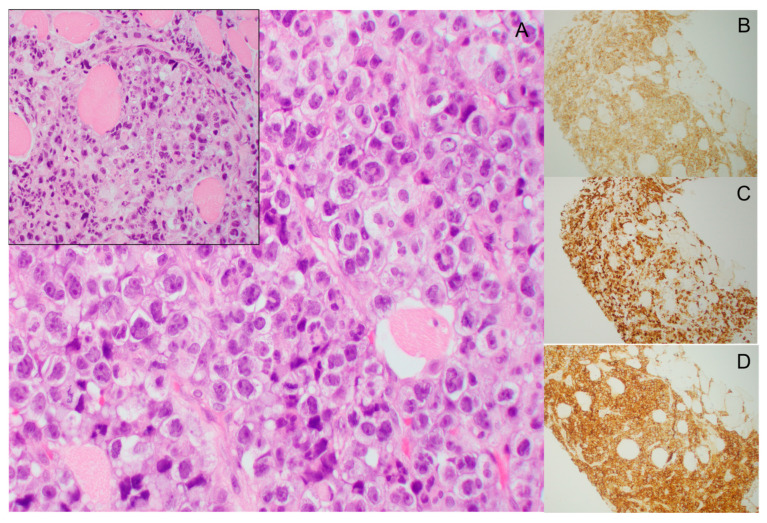
Histology of myeloid sarcoma involving the psoas muscle. (**A**, 500×). The tumor cells are blasts with irregular/convoluted nuclei, finely dispersed chromatin, and small distinct nucleoli. The inset (400×) shows the tumor cells infiltrating between skeletal muscle fibers. The tumor cells are immunoreactive for CD33 (**B**, 100×), CD163 (**C**, 100×), and CD45 (**D**, 100×).

**Figure 3 cancers-15-01443-f003:**
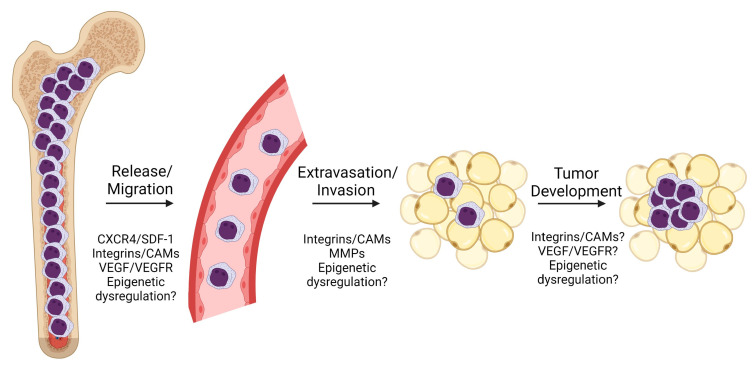
Depiction of MS development in AML with proposed mediators. Initial release/migration of leukemic blasts from the bone marrow into the peripheral blood circulation, followed by extravasation and invasion into distant tissue spaces (e.g., subcutaneous tissue). This results in organization of those extravasated cells into a mass with subsequent tissue architectural distortion and gross observation and clinical symptoms. Abbreviations: CAMs, cell adhesion molecules; MMPs, matrix metalloproteinases; VEGF, vascular endothelial growth factor; VEGFR, vascular endothelial growth factor receptor. Figure created with BioRender.com.

**Table 1 cancers-15-01443-t001:** Summary of survival outcomes of pediatric patients with AML and extramedullary disease. Described terminology as per original report. Abbreviations: YO, years old; EMD, extramedullary disease; CSF, cerebral spinal fluid; EML, extramedullary leukemia; EMI, extramedullary infiltration; MS, myeloid sarcoma; EFS, event free survival; OS, overall survival; RFS, relapse free survival; SE, standard error; AML, acute myeloid leukemia; CI, confidence interval [10,11,12,13,14,15,16,17,18,19,20,21,22,23].

Study/Publication	Age	Extramedullary Disease Involvement Study Definitions	Population	Incidence	5-Year Estimated EFS (±SE) or (95% CI)	5-Year Estimated OS (±SE) or (95% CI)
POG8821 (Chang, et. al., 2000) [10]	<21 yo	EMD: including CSF disease, not defined	n = 492	Any EMD 10.4%	**4-year EFS**		Not available	
				CSF only 4.7%	CSF only: 34.8 ± 9.9%	*p* = 0.91		
				Non-CSF EMD 5.7%	Non-CSF EMD: 21.6 ± 8.6%	*p* = 0.043		
					No EMD: 34.4 ± 2.5%	*p* = 0.18		
DCLSG (Bisschop et. al., 2001) [11]	0–16 yo	EML: Clinically obvious infiltrate in soft tissues, skin, muscles or bone, gingiva, CSF or brain	n = 477	EML in 25.1%	No EML 38% ± 3%	*p* = 0.85	Not available	
					Myeloblastoma (MS) 43 ± 13%			
					Skin infiltrates 45 ± 21%			
Children’s Cancer Group, CCG AML 213 and 213P, 2861 and 2891 (Dusenbery et. al., 2003) [12]	0–21 yo	“Chloroma” on data entry form yes or no, gum only not included	n = 1832	Skin EML ± other 5.9%	Skin ± other: 26% (17–35%)	*p* = 0.005	Not available	
		“Skin involvement” yes or no		Non skin EML 4.9%	Non skin EML: 46% (34–58%)			
				EML 10.9%	Non EML: 29% (27–32%)			
Single Center—Turkey (AML-90 and AML-94 protocols) (Hiçsönmez et. al., 2004) [13]	<17 yo	EMI: involvement of gingiva, CNS, orbit, soft tissue, bone, pleura	n = 127	EMI total in 40%	**4-year EFS:**		Not available	
				Gingiva only in 11%	AML-90 therapy: MS = 0%	*p* < 0.05		
				Orbital in 10%	Without EMI = 37 ± 11%			
				MS in 21%	AML-94 therapy: MS = 56 ± 17%	*p* > 0.05		
					Without EMI = 31 ± 1%			
Japanese childhood AML cooperative study group (Kobayashi et. al., 2007) [14]	<16 yo	CNS disease (>5 WBC/μL with blasts)	n = 240	EMI in 23.3%	**3-year estimate EFS**		**3-year OS**	
		EMI: leukemic infiltration in organs other than liver, spleen, lymph nodes (including CNS disease)		(Excluding CSF only: 20.4%)	EMI: 53.3 ± 6.7%	*p* = 0.11	EMI: 77.3%	
					No EMI: 62.5 ± 3.6%		No EMI: 77.6%	
					EMI + WBC > 100 × 10^9^/L: 23.8 ± 12.9%	*p* = 0.0052		
					No EMI or EMI + WBC < 100 × 10^9^/L: 60 ± 3.5%			
Children’s Oncology Group (CCG 2861, 2891, 2941, 2961) (Johnston et. al., 2012) [15]	0–21 yo	CNS3 (≥5 WBC/μL with blasts)	n = 1459	CNS3 11%	No MS 40 ± 3%	*p* = 0.005	No MS 50 ± 3%	*p* < 0.001
		CNS MS (brain or spinal cord tumor)		CNS MS 1%	CNS MS 52 ± 21%		CNS MS 73 ± 19%	
				Orbital MS 2%	Orbital MS 76 ± 17%		Orbital MS 92 ± 11%	
				Non CNS MS 4%	Non CNS MS 34 ± 13%		Non CNS MS 38 ± 13%	
European AML Study Groups (Creutzig et. al., 2017) [16]	0–17 yo	CNS involvement	n = 2365	CNS 11.0%	CNS + 48 ± 3%	*p* = 0.11	CNS + 64 ± 3%	*p* = 0.23
		(CSF with >5 WBC/μL with blasts or intracranial infiltrates on imaging or neurologic symptoms)			CNS—52 ± 2%		CNS—67 ± 1%	
NOPHO AML 2004 (Støve et. al., 2017) [17]	0–17 yo	MS: myeloblast tumor	n = 322	MS (± CNS disease) 15.8%	EML: 54% (42–65%)	*p* = 0.57		*p* = 0.008
		CNS disease (≥ 5 WBC/μL with blasts or new neurologic symptoms)		CNS only an additional 7%	No EML: 45% (37–51%)		EML: 64% (51–74%)	
		EML: MS or CNS disease					No EML: 73% (66–78%)	
Single Center—India (Pramanik et. al., 2018) [18]	0–18 yo	MS (did not include CSF only disease)	n = 570	MS in 21.2%	**Median EFS:**	*p* = 0.002	**Median OS:**	*p* = 0.002
					AML with MS: 21.6 months		With MS: 26.3 months	
					AML without MS: 11.1 months		Without MS: 12.7 months	
TARGET dataset (COG-NCI) (COG AAML03P1, AAML0531, CCG-2961) (Xu et. al., 2020) [19]	<18 yo	MS on biopsy diagnosis, excluding CSF disease	n = 884	MS in 12.3%	MS: 35.4 ± 4.6%	*p* = 0.001	MS: 53.4 ± 4.8%	*p* = 0.008
					Non-MS: 48.5 ± 1.8%		Non-MS: 64.0 ± 1.8%	
Single Center—Korea (Lee et. al., 2020) [20]	<18 yo	EMI: excluded CSF only	n = 40	EMI in 30%	EMI: 50.0 ± 14.4%	*p* = 0.022	Not available	
**Only *RUNX1-RUNX1T1* AML**					No EMI: 78.6 ± 7.8%			
Single Center—China (Hu et. al., 2020) [21]	≤18 yo	MS: including lymph nodes >2cm, excluded CNSL	n = 214	MS in 20.6%	**3-year RFS**	*p* = 0.000	**3-year OS**	*p* = 0.01
**Only Low Risk AML (includes Hu et. al., 2021 study)**					With MS: 62.6 ± 7.5%		With MS 73.5 ± 7.1%	
					Without MS: 87.0 ± 2.8%		Without MS 88.8 ± 2.6%	
Single Center—China (Hu et. al., 2021) [22]	1–18 y	MS: clinical, biopsy, radiology findings	n = 127	MS in 23.6%	**3-year RFS**	*p* = 0.004	**3-year OS**	*p* = 0.249
**Only t(8;21) AML**		CNS MS: dura deposits or paraspinal tumor			With MS: 68.8 ± 8.8%		With MS: 78.1 ± 8.1%	
	o				Without MS: 88.0 ± 3.4%		Without MS: 86.4 ± 3.7%	
Polish Pediatric Leukemia and Lymphoma Study Group (Samborska et. al., 2022) [23]	0–18 yo	MS: pathology diagnosis or extramedullary tumor and concurrent bone marrow disease (AML, MDS)	n = 43	MS in 100%	De novo: 0.56 ± 0.12	*p* = 0.0247	**pOS**	*p* = 0.0251
				De novo/isolated in 37.2%	Concurrent: 0.82 ± 0.08		De novo: 0.56 ± 0.12	
				Concurrent in 55.8%			Concurrent: 0.84 ± 0.09

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
