# Peer review of "Pediatric Myeloid Sarcoma, More than Just a Chloroma: A Review of Clinical Presentations, Significance, and Biology"

_cancers, 2023, doi:10.3390/cancers15051443_

Round 1

Reviewer 1 Report

Zorn KE's text is very well detailed and interesting. It describes the clinical, radiological and biological characteristics of myeloid sarcoma. The introductory part is suffciently concise. The clinical section describes in detail the presenting characteristics of the disease and is enriched with a summary table which also reports the probability of survival and the EFS. The treatment section proposes aspects derived from SM biology that appear interesting and worthy of further study. A summary table with the main outcome results according to the treatment could be useful: HSCT, hypomethylating drugs etc. The biology and pathogenesis section, although related to possible treatments, should be placed before the treatment chapter. There are typos that need to be corrected throughout the text

Author Response

Point 1: The treatment section proposes aspects derived from SM biology that appear interesting and worthy of further study. A summary table with the main outcome results according to the treatment could be useful: HSCT, hypomethylating drugs etc.

Response 1: We appreciate this suggestion from the reviewer.  Unfortunately, when reviewing the studies, the patient groups were too heterogenous for there to be any discernible outcomes, and in fact different outcomes with different therapies were not always clearly delineated.  As such, we were unable to find an effective way to summarize the treatment approaches as requested by the reviewer. We apologize that this is not possible, but would like to note for the reader who is truly interested the references can be used as a starting point.

Point 2: The biology and pathogenesis section, although related to possible treatments, should be placed before the treatment chapter.

Response 2: Thank you for this suggestion. We will reorder the manuscript accordingly.

Point 3: There are typos that need to be corrected throughout the text

Response 3: Thank you for this concern. We will carefully review and correct any typos and errors.

Reviewer 2 Report

The manuscript by Zorn et al is a well written review of what is known about myeloid sarcomas in both pediatric and adult patients.  Data on incidence across a multitude of study groups as well as outcome data is presented.  Perhaps most compelling clinical piece of information presented is the synthesis of data around the use of allo-HSCT and the apparent lack of improvement in outcome for individuals with MS who show evidence of GvHD – which is frequently used as a proxy for GvL effect. The review also has a well written synopsis of what is currently known about different potential biologic drivers of MS.  In Summary this is a well written review article on a poorly understood disease processes within the relatively rare population of pediatric AML patients.   

Minor comments;

Simple summary line 18: change improve to improving

Line 417-418 – sentence is unclear – are the authors trying to state that imaging for MS is inconsistent?  The way it currently reads is that the challenges are inconsistent.  Would remove word inconsistent, or could change to “use of imaging, both frequency and modality, to identify occult MS … have remained inconsistent, particularly in children.”

Author Response

Point 1: Recommend Simple summary line 18: change improve to improving

Response 1: We appreciate the reviewer’s suggestion and will incorporate this change into the manuscript.

Point 2: Line 417-418 – sentence is unclear – are the authors trying to state that imaging for MS is inconsistent?  The way it currently reads is that the challenges are inconsistent.  Would remove word inconsistent, or could change to “use of imaging, both frequency and modality, to identify occult MS … have remained inconsistent, particularly in children.”

Response 2: We appreciate the reviewer’s suggestion and will incorporate this change into the manuscript for better clarity. The use of inconsistency was in reference to the imaging not the challenges as was mentioned above.

Reviewer 3 Report

It is very interesting review of current knowledge of myeloid sarcoma in children. Authors discuss the clinical presentation, diagnostics, therapeutic approach, outcome and finally the biology of pediatric patients with MS.

The review is very well organized and written. As MS in children is still inadequately understood this work is of great significance.

I have 2 minor comments:

Line 293-294

Current pediatric AML protocols do not include radiotherapy as standard treatment of MS -  radiotherapy is still recommended for extramedullary changes in AML-BFM recommendations 2019.

Authors refer mainly studies on pediatric MS from North America or Asia. Only a few studies of European study groups are cited. I propose to add the recent retrospective study of Polish Pediatric Leukemia Lymphoma Study Group concerning myeloid sarcoma.

Author Response

Point 1: Current pediatric AML protocols do not include radiotherapy as standard treatment of MS -  radiotherapy is still recommended for extramedullary changes in AML-BFM recommendations 2019.

Response 1: Thank you for raising this concern. Our statement emphasized current Children Oncology Group (COG) standard of care and we apologize if this made the review seem exclusionary, especially for important perspectives from Europe. We propose the change below. We welcome any references otherwise regarding BFM standards as we are limited in our ability to access and yet seek accuracy in our statement.

"Although prior Children’s Oncology Group (COG) studies included radiotherapy for treatment of MS sarcomas this is no longer standard of care in the US; however, it continues to be recommended for MS in Berlin-Frankfurt-Munster (BFM) studies."

We have also added (Creutzig et al 2012 AML Committee of the International BFM Study Group. Diagnosis and management of acute myeloid leukemia in children and adolescents: recommendations from an international expert panel) which includes details on a prior BFM based study.

Point 2: Authors refer mainly studies on pediatric MS from North America or Asia. Only a few studies of European study groups are cited. I propose to add the recent retrospective study of Polish Pediatric Leukemia Lymphoma Study Group concerning myeloid sarcoma.

Response 2: Thank you for this suggestion. We did not mean to exclude this article and will include this study in the updated manuscript particularly with regards to the described outcomes and patient characteristics.